# The Clinical Management of Pompe Disease: A Pediatric Perspective

**DOI:** 10.3390/children9091404

**Published:** 2022-09-16

**Authors:** Jorge Sales Marques

**Affiliations:** 1Conde S. Januário Hospital, Macau 999078, China; jorge.marques@jmellosaude.pt; 2Hospital Cuf Trindade, 4000-541 Porto, Portugal

**Keywords:** Pompe disease, newborn screening, alglucosidase alpha, enzyme replacement therapy, clinical management

## Abstract

Pompe disease (PD) is an inherited metabolic disorder caused by a deficiency of acid α-glucosidase (GAA), leading to lysosomal accumulation of glycogen, mainly in skeletal and cardiac muscles as well as the nervous system. Patients with PD develop cellular dysfunction and muscle damage. PD can be classified into two classic forms, namely infantile-onset PD (IOPD) and late-onset PD (LOPD). Delayed treatment, particularly in IOPD, would result in significant organ damage and early death. Nonetheless, early diagnosis and timely treatment are often hampered by the rarity of PD and its wide variety of, but overlapping, symptoms. This article reviews the common clinical presentations of PD and outlines the essentials of PD management. In particular, the implications of newborn screening (NBS) and clinical performance of enzyme replacement therapy (ERT) are highlighted.

## 1. Introduction

Pompe disease (PD), which is also referred to as acid maltase deficiency or glycogen storage disease type II, is a metabolic disorder triggered by biallelic gene mutations in chromosome 17q25, which encodes for acid α-glucosidase (GAA), a lysosomal hydrolase, leading to a deficiency of the enzyme and, hence, lysosomal accumulation of glycogen, particularly in skeletal and cardiac muscle as well as the nervous system. This leads to cellular dysfunction and muscle damage [1]. The severity of PD is determined by the level of residual GAA activity; thus, the disease can be controlled by enzyme replacement therapy (ERT) [2]. The incidence of PD is estimated to be 1 in 40,000 live births, varying based on ethnicity and geographical regions. For instance, the reported incidences in Austria and Taiwan were 1 in 8684 and 1 in 17,000, respectively, which are much higher than the global estimate [1].

## 2. IOPD and LOPD

The phenotypic variation of PD is broad, and the age of symptom onset varies from early infancy to adulthood. Hence, two classic forms of the disease are recognized in the literature, namely infantile-onset PD (IOPD) and late-onset PD (LOPD). Notably, LOPD can be further categorized into childhood, juvenile, and adult-onset PD [3]. Moreover, patients with residual GAA are classified as cross-reactive immunologic material (CRIM)-positive while those that lack the enzyme completely are CRIM-negative [4]. In IOPD, GAA activity is almost completely absent, normally <1% of the mean activity as compared to healthy individuals, resulting in rapidly progressive muscle weakness, hypotonia, macroglossia, hepatomegaly, and progressive hypertrophic cardiomyopathy (HCM) [5]. In addition, feeding difficulties and respiratory tract infections have been reported in patients with IOPD as well [6]. Essentially, the symptoms of IOPD typically onset within the initial few weeks of life, and death because of cardiorespiratory failure would occur within 12 months if untreated [7].

Patients with LOPD have pathogenic GAA variants which do not abolish GAA activity completely. Thus, instead of early onset during infancy, the age of onset of LOPD can range from less than 1 years old to 50s [8]. Besides the age of onset, LOPD is generally variable in disease progression and clinical manifestation as well. Skeletal muscle dysfunction that developed in LOPD would lead to progressive respiratory insufficiency and limb–girdle weakness [1].

## 3. Clinical Presentations of Pompe Disease

The impact of PD depends mainly on the level of residual GAA activity, where lower residual activity results in earlier onset and more aggressive disease with worse prognosis [9]. Notably, the onset age does not always reflect subtypes of PD well, where late-infantile or non-classic PD may manifest during infancy as well [2]. The variety of overlapping clinical presentations of PD hence hinders timely diagnosis and initiation of treatment.

Symptoms of IOPD usually develop within the initial few months of life, with a median age of 2.0 months and diagnosis at 4.7 months. The reported 12-month overall survival rate was 25.7%, whereas ventilator-free survival was 16.9%. Early symptom onset was demonstrated to increase the risk of early death, mainly because of cardiorespiratory failure [10]. A former investigation revealed that symptoms of IOPD appear after a median age of about 4.0 months [10]. Recently, a cohort study by Dorpel et al. (2020) involving 16 classic infantile patients reflected that the majority (88%) of the patients exhibited distal muscle weakness of the lower extremities, while 31% of them developed weakness of the hands [11]. These findings suggested that distal muscle weakness would precede the development of proximal muscle weakness.

The phenotype of PD patients diagnosed beyond the first year of life can be varied from asymptomatic with elevated creatine kinase (CK) to distinct muscle weakness. Cardiac involvement is typically rare in LOPD, whereas symptoms of LOPD are mainly related to progressive skeletal muscle dysfunction initiated in the proximal lower limb and paraspinal trunk muscles. Essentially, respiratory failure associated with the dysfunction of the diaphragm and accessory muscles of respiration is a major cause of morbidity and mortality in LOPD [3]. A brief summary of common clinical presentations of PD is listed in Table 1.

## 4. Examinations and Diagnosis of Pompe Disease

Although timely diagnosis and prompt treatment are vital for improving patients’ survival, especially for IOPD, delayed diagnosis of PD is common. It has been reported that the diagnostic gap is shorter in patients with classic IOPD, but significant delays were observed in patients with LOPD [12]. As stated, most of the undiagnosed or untreated IOPD patients will die of disease progression in the first year of life [3]. The rarity of the disease as well as its highly diversified but overlapping clinical presentations are some of the factors hampering early identification of PD cases. Given the significant negative outcomes of IOPD, classic infantile disease has to be differentiated from the non-classic/late-infantile PD at the initial stage of diagnosis. Summarizing the information from former reports, major symptoms and clinical observations suggestive of potential PD are highlighted in Table 2.

Upon detection of symptoms suggestive of PD, further clinical assessments are required to document a PD diagnosis because of the varied and non-specific nature of clinical presentations. As PD is associated mainly with muscular, respiratory, neurophysiological, and cardiac systems, clinical examinations on these organ systems and biochemical tests are included in the diagnostic protocol.

### 4.1. Muscular System

Muscle biopsy is a diagnostic tool used in the evaluation of muscle diseases. In IOPD, periodic acid Schiff (PAS) positivity, cytoplasmic vacuoles [14], and reactive to acid phosphatase [15] are histological signs suggesting PD diagnosis. However, PAS-positive vacuolar myopathy alone for identifying PD often leads to false-negative results [16]. In evaluating the ultrastructure, extra- and intra-lysosomal glycogen, autophagic vacuoles and structure distortion with a marked reduction in myofibrils could be observed [17]. The biopsy observations in LOPD might appear normal or similar to IOPD [17]. Of importance, biopsy results have to be confirmed with either a DNA test or determination of the GAA activity. Moreover, in view of the traumatic nature of muscle biopsy, less invasive options, such as the blood spot test, are preferred.

### 4.2. Respiratory System

Dysfunction of respiratory muscles is commonly observed in PD, whereas the condition can be developed before the involvement of limb and other muscles. Thus, an examination of respiratory function is recommended. Clinically, a postural change in measurement of forced vital capacity (FVC), from upright to supine, of >20% would indicate diaphragmatic weakness. Moreover, PD patients would exhibit substantial drops in vital capacity from seated to supine position [18]. As respiratory muscle weakness would increase PD patients’ risk of fatal respiratory tract infections, the maximum expiratory pressure (MEP) of the patients should be measured [18]. Furthermore, plethysmographic assessment of lung volumes is recommended to reflect the changes in vital capacity and residual volume, whereas evaluating the maximal respiratory pressures would help detect the occurrence of respiratory muscle weakness [19]. 

Early diagnosis of airway abnormalities is important for ensuring better outcomes. Thus, airway examination through flexible bronchoscopy (FB) is recommended for PD patients. In clinical practice in Taiwan, polysomnography is recommended for PD patients from the age of 6 months, whereas pulmonary function tests should be performed from 3 years old with follow up every six months [20]. With more reliable information of respiratory conditions obtained from FB and the functional tests, earlier interventions, such as higher ERT and nighttime nasal intermittent positive pressure ventilation, would be justified.

### 4.3. Neurophysiological System

In patients with LOPD, nerve conduction studies (NCS) are normal, whereas electromyography (EMG) frequently, approximately 70% [21], shows myopathy with increased muscle membrane irritability [22]. Notably, EMG abnormalities can be observed in paraspinal muscles only, with limb muscles appearing as normal [23]. In IOPD, in addition to membrane irritability and myotonic discharges, the voluntary contraction would show short, low-amplitude, and polyphasic motor unit action potentials [17].

### 4.4. Cardiac System

Hypertrophic cardiomyopathy is a major feature indicating cardiac involvement in patients with IOPD. Hence, chest X-ray, electrocardiography (ECG) and echocardiogram can be applied to guide diagnosis. For instance, a chest X-ray may reflect cardiomegaly, reduction in lung volume and/or regions of atelectasis in IOPD, but would appear as normal in LOPD [17]. In addition, Ansong et al. (2006) reported that a short PR interval and tall QRS complexes were observed in the ECG of about 75% of patients with IOPD [24]. Furthermore, an ECG would reveal hypertrophic cardiomyopathy in the early stages of IOPD [25].

### 4.5. Biochemical Tests

Analysis of GAA activity in fibroblast cultures obtained from skin biopsy can be applied in PD diagnosis [17,26]. In patients with PD, reported residual GAA activity varies from 2% to 40% of normal activity [3]. However, due to the long turnaround time, technical difficulties of fibroblast culture, the dried blood spot (DBS) test is adopted in the PD diagnosis because of its speed and convenience [27]. The DBS test is non-invasive with high sensitivity and specificity in identifying PD individuals [28]. Notably, previous evaluation with mass spectrometry indicated that no significant difference exists between blood and fibroblast measurements [29]. Glucose tetrasaccharide (Glc4) is a urinary biomarker of GAA activity for evaluating treatment outcome. Elevation of Glc4 levels in urine has been reported in patients with IOPD, with sensitivity close to 100% [30]. 

Muscle damage would lead to an increase in CK, which is a sensitive marker for PD. Increased CK levels are observed in all IOPD and the majority of LOPD patients, with levels ranging from 1.5 to 15 times the upper limit of normal [22]. Nonetheless, it is essential to be aware that normal CK levels can be observed in LOPD as well [31].

## 5. The Importance of Newborn Screening for PD

ERT is essential in the management of PD, IOPD in particular, and ERT should be started before the onset of symptoms to avoid irreversible damage and, hence, to achieve optimal outcomes [32]. Practically, timely diagnosis of PD and subsequent treatment are hampered because of the lack of screening tests and limited awareness of the rare disease amongst healthcare professionals [33]. While many children with PD are not diagnosed until after referral to a clinical specialist, it is sad but true that many of the children do not survive the time it takes to confirm the diagnosis. This hence highlights the importance of newborn screening (NBS) for PD.

### 5.1. Global Practice on Pompe NBS 

The first pilot study on NBS for PD was conducted in 2005 in Taiwan. In the screening, two-tier DBS analyses were performed to assess GAA activity [34]. The pilot NBS on 473,738 newborn samples identified 9 classic IOPD cases and 19 LOPD cases, including non-classic IOPD [35]. Thus, the pilot NBS in Taiwan demonstrated that the practice not only facilitates timely treatment for IOPD but also allows detection of underdiagnosed LOPD and asymptomatic GAA-deficient individuals [36].

More recently, Sawada and co-workers (2021) reported findings in Pompe NBS in Japan. In the study, among the 296,759 newborns measured, one IOPD and seven LOPD patients were identified. In addition, 34 pseudodeficient individuals and 65 carriers or potential carriers were found. Importantly, the IOPD patients identified were prescribed early ERT before presenting exacerbated manifestations [37].

In the United States, currently, more than 20 states have implemented NBS for PD. Differing from the pilot NBS in Taiwan, tandem mass spectrometry (TMS) compared to digital microfluidics (DMF), with additional reflex testing on initial NBS samples, was used for measurement of primary analytes during PD screening in most of the states. While some other states reported planning for start screening, NBS for PD is expected to be implemented in more states [38]. 

### 5.2. Next Generation Sequencing and Mutation Analysis

While differential diagnosis of PD may be difficult based solely on clinical presentations, next-generation sequencing (NGS) can test concurrently for multiple genetic muscle disorders. Currently, >500 sequence feature variants in the GAA gene have been identified, with more than 350 displaying pathogenicity [17]. Although analysis of GAA gene mutation is less specific, the technology is useful in detecting carriers when a familial mutation is identified [3].

Certain common mutations can be applied to show the genotype–phenotype correlations. For instance, it is reported that the variant p.Arg854Ter was found in about 50–60% of African Americans with IOPD [39], whereas 50–85% of adults with LOPD have the variant c.336-13T > G [40,41]. For LOPD in Chinese patients, Liu et al. (2014) demonstrated that c.2238G > C (p.W746C) is the most common mutation observed [42].

### 5.3. The Concern on NBS in Pompe Disease

Though NBS would facilitate early detection and treatment for PD patients, it is not without limitations. The occurrence of pseudodeficiency can complicate NBS in PD, particularly in Asian populations. The reported prevalence of homozygous pseudodeficiency alleles in Taiwan and Japan was 3.3% and 3.9%, respectively [43]. Nonetheless, optimization of the testing protocol would facilitate better evaluation of a patient’s disease condition. 

### 5.4. Clinical Benefits of Pompe NBS-Guided Early Treatment

Timing of treatment initiation is a critical factor determining the morbidity and mortality of patients with PD, whereas the clinical data from global NBS programs illustrated the improved outcomes yielded by NBS-guided early diagnosis followed by timely treatment. Chien et al. (2009) conducted an NBS pilot program, which involved 206,088 newborns. Six infants tested positive for PD and five of them developed rapid progression of PD as reflected by the motor and cardiac involvement. These patients were prescribed with intravenous alglucosidase alpha upon diagnosis. The sixth patient initiated treatment upon onset of progressive muscle weakness at 14 months of age [44]. 

With early alglucosidase alpha treatment, the patients exhibited normal physical growth and age-appropriate gain in motor development. Importantly, overall survival and invasive ventilator-free survival were 100% in the NBS group, which was significantly better than the untreated cohort (*p* = 0.001) [44]. The results thus confirmed that NBS-guided early treatment for PD would significantly improve patient outcomes.

## 6. ERT for PD

ERT with recombinant human GAA (rhGAA; Myozyme^®^ (Sanofi, Cambridge, MA, USA), alglucosidase alpha) is a treatment available for PD. ERT can effectively reverse the cardiomyopathy and improve the clinical course as well as the expected outcomes of PD patients [44,45,46]. Nonetheless, treatment may not completely resolve the symptoms in patients who initiated the treatment after five months of age, or in those with a marked increase in left ventricular mass index (LVMI) [47]. Early initiation of ERT is thus crucial for improving survival, reducing the need for ventilation, facilitating earlier independent walking, and enhancing patients’ quality of life [44]. 

In prescribing ERT with alglucosidase alpha, a dosage of 20 mg/kg body weight biweekly is recommended. The treatment is generally well-tolerated; however, significant hypersensitivity reactions were reported in less than 1% of patients [48].

Most ERT-associated adverse events (AEs) are mild to moderate, transient infusion-associated reactions (IARs), where patients may continue treatment without medical treatment or discontinuation of infusion. IARs may occur during and up to 3 h after infusion [26] and usually stop after discontinuation of infusion but may reappear upon subsequent infusion. In cases of anaphylaxis or severe allergic reactions, reducing the infusion time and immediate termination of administration should be considered and a pre-treatment dose of oral antihistamine and/or antipyretics and/or corticosteroids can be applied [48].

### Dose Optimization in ERT

While very early ERT for patients with PD guided by NBS yielded favorable outcomes, clinical studies suggested that further benefits would be generated with a higher dosage and higher frequency of ERT [49]. In a previous trial by Gelder et al. (2016) involving eight IOPD patients, four of them were prescribed with alglucosidase alpha with a dose of 20 mg/kg every other week (standard dose) and four were treated weekly at a higher dose of 40 mg/kg. The results demonstrated that all patients survived at the end of the study. One of the patients on a standard dose became ventilator-dependent, while the remaining seven were ventilator-free. All four patients on a higher dose could walk at the end of the study, whereas only two maintained walking ability in the standard dose group. Notably, the baseline motor functioning of patients in the high-dose group was poorer. Nonetheless, no significant differences in LVMI, IARs and antibody formation were observed between two groups [50]. According to the results, a higher dose of ERT would likely improve outcomes in PD patients as compared to the standard dosage.

More recently, a retrospective study by Chien et al. (2020) involving 28 IOPD patients on high-dose ERT (40 mg/kg biweekly) reported that patients who were late in ERT initiation (*p* = 0.006) or late in high-dose ERT initiation (*p* = 0.044) were at a higher risk of motor decline. Furthermore, serum CK level and urinary Glc4 level were correlated with the favorable response to ERT in the patients. The results indicated that prescription of a high-dose ERT immediately upon positive findings at NBS gave the best outcomes, and a dosage increase is needed upon a rise in biomarker levels [51].

## 7. Recommended Treatment Protocol for PD

While ERT is the main component in the management of PD, general care measures are crucial. Therefore, a multidisciplinary team of healthcare professionals, including cardiologists, occupational therapists, nurses, etc., has to be involved in the management of PD. For instance, respiratory muscle weakness is one of the major health problems faced by patients with PD, particularly IOPD, despite being treated with ERT. An intensive respiratory muscle training (RMT) regimen has been demonstrated to be beneficial for improving inspiratory and expiratory muscle strength in PD survivors [52].

Although impacts on cardiac and skeletal muscles are the most important in PD, the prevalence of sensorineural hearing defects is high among patients with classic IOPD [53]. Thus, close monitoring of auditory function is needed for classic IOPD patients. Timely intervention with hearing aids is essential for speech and language development.

Scoliosis is common in many neuromuscular disorders, whereas data from the Pompe Registry suggest that one-third of PD patients suffer from scoliosis. The condition was more common in patients with PD symptoms onset as children and juveniles than in those who had onset as adults [54]. Scoliosis is associated with a higher risk of clinical morbidity and, hence, assessment of scoliosis in all PD patients is desirable. Practically, supportive products should be used if needed.

### 7.1. The Importance of CRIM Status and Management of CRIM-Negative Patients

While ERT should be initiated as soon as the diagnosis of IOPD or symptomatic PD is established, determining the patients’ CRIM status before initiating treatment is recommended. Of importance, CRIM-negative individuals may develop resistance against rhGAA during ERT, and modified therapy protocols using immunomodulation may be required [7].

There are several tests available for determining CRIM status including Western blot analysis of skin fibroblasts. Given the invasiveness and long duration of cultured skin fibroblasts, a less-invasive blood-based CRIM assay was developed to shorten the time required [55]. Moreover, CRIM status can also be predicted by GAA variant analysis if the pathogenic variants are already identified [4]. Clinical opinion suggests that immune tolerance induction (ITI) with medications such as methotrexate, rituximab and intravenous immunoglobulin (IVIG) with/without bortezomib at ERT commencement would optimize immune tolerance to ERT in CRIM-negative patients [56].

### 7.2. ERT in Practice

Before ERT, patients and their families have to be informed about the treatment goals. During the course of ERT, the occurrence of any adverse event, no matter if it is related to ERT or not, should always be recorded. Essentially, patients receiving ERT should be monitored for the development of AEs involving skin and other organs. In cases of severe allergic reactions, termination of ERT might be considered with the advice of a team of healthcare professionals. In addition, ERT is not recommended for patients who suffer from severe infusion-associated reactions that cannot be properly managed, with the presence of other life-threatening disease, and those with another medical condition that might compromise the response to ERT [57]. Of note, neurocognitive problems in IOPD cannot be managed by ERT with rhGAA [2]. As a rule of thumb, the involvement of patients’ families in the decision process, including discontinuation of ERT treatment, in PD management is vital. Upon a diagnosis of classic IOPD, CRIM-positive status is documented; excluding cases of being carriers or pseudodeficiency, ERT should be commenced. 

In patients with LOPD, further evidence indicating the appropriate time to start ERT in asymptomatic patients is still needed. Previous trials demonstrated that improved outcomes of ERT can be achieved when treatment was initiated upon detection of the first measurable signs of disease and/or subtle symptoms though still considered as clinically asymptomatic [58]. Thus, current consensus advocates the initiation of ERT for asymptomatic LOPD patients without objective clinical signs upon symptoms appearing and the onset of proximal muscle weakness and/or respiratory involvement with a >10% drop in sitting-supine FVC [59]. Furthermore, in clinically asymptomatic patients, ERT should be considered when muscle weakness is detectable through directed examination [59]. 

For LOPD patients with objective signs and/or symptoms present, ERT should start upon confirmation of the positive clinical test results with the consensus of the multidisciplinary team and individual patients and/or families [4]. In severe cases, ERT should be continued if there is stabilization or improvement of severe signs and symptoms [59]. A flowchart summarizing the treatment protocol of ERT in PD is illustrated in Figure 1.

### 7.3. Monitoring of ERT and Follow-Up

Monitoring of ERT treatment outcomes and disease progression status is essential in the management of PD, particularly for LOPD patients. It is recommended for the patients to be assessed at the same time of day by the same examiner to reduce the impacts of confounding variables [26]. To evaluate the outcomes of ERT therapy, IgG antibodies of patients should be monitored every 3 months for 2 years, then annually thereafter. The effectiveness of ERT has to be assessed annually in order to evaluate whether continuing ERT is warranted [59]. Furthermore, follow-up assessments for PD patients include manual muscle testing on skeletal muscle strength, sitting-supine FVC, 6 min walk test indicating functional exercise capacity, quality of life (QoL) measurement, audiology assessment and laboratory tests on CK level and GAA antibody titers. It is recommended that the follow-up assessments for LOPD patients undergoing ERT should be conducted every 6 months [26]. A summary of assessments for monitoring PD disease progression is listed in Table 3.

In cases where no relief of symptoms is observed or, even worse, there is disease progression despite regular ERT, termination of ERT should be considered. Moreover, failure to comply with ERT treatment or measures adequately, or the development of life-threatening complications and/or another disease with poor prognosis, would be the signs for terminating ERT as well.

### 7.4. Genetic Counselling

As PD is an inherited disease, genetic counseling for all parents with an affected child and all adults with PD is essential. Typically, genetic counseling of PD can be focused on prenatal testing and early diagnosis in the newborn. In interpreting the positive Pompe NBS resulting in patients’ parents during genetic counselling, some important rules have to be followed. Firstly, it is not appropriate to minimize the initial results for parents with the possibility of false-positive results. It is also not appropriate to tell parents that IOPD implies the child will die in infancy. Moreover, it is not advisable to ask the parents not to check information about PD from online sources but to give them suggestions on checking valid and reliable sources and preventing incorrect information. It is crucial to deliver the message that a baby with a healthy outlook can still inherit a genetic disease. For instance, the common c.-32-13T > G variant of PD would not result in IOPD but predicts LOPD [8].

## 8. Case Sharing on ERT for PD

To illustrate the efficacy of ERT with alglucosidase alpha in PD in real life settings, the clinical cases of five patients (four cases of classic IOPD and one LOPD case) are presented in the current article. All four IOPD cases occurred in male patients aged 3 (Patient A), 4 (Patient B), 7 (Patient C), and 10 (Patient D) months old, respectively. All four patients exhibited hypotonia and were subsequently found to have elevated CK of 40% above normal range. In particular, hypertrophic cardiomyopathy was detected in Patient D at 4 months of age. All of the patients were treated with alglucosidase alpha 20 mg/kg body weight administered weekly. All of the patients, including Patient D, responded well to the ERT and achieved improved muscle tone with reduced CK after 6 months of double dose treatment.

The remaining case (Patient E) was a 2-year-old boy with LOPD. The patient presented with weakness and difficulty in climbing. Laboratory tests revealed that the patient’s CK was only 10% of normal range value. Hence, weekly treatment of alglucosidase alpha 20 mg/kg body weight was prescribed. After 6 months of ERT, normalization of CK level to the normal range was achieved and the patient recovered from the weakness to the extent that he was able to climb up stairs. The efficacy of ERT was sustainable, where all five patients recovered well in follow-up, with no treatment-related adverse events reported. Thus, the treatment dosage was not changed.

## 9. Perspectives in Pompe Disease Management

ERT with alglucosidase alpha substantially improves the survival and wellbeing of patients with PD. Subsequent investigations on physiological responses to the treatment allow a better understanding of the pathophysiology of PD and hence provided insights on further development on new therapeutics for the disease. For instance, alglucosidase alpha has been reported to clear glycogen storage in cardiac muscle more effectively than in skeletal muscle, which partially reflects the tissue differences in cation-independent mannose 6-phosphate (CIM6P)/insulin growth factor II (IGF-II) receptor expression and enzyme uptake [61]. While the cell-surface CIM6P/IGFII receptor mediates cellular uptake of exogenous GAA and targets it to the lysosomal compartment [62], increasing M6P on the recombinant enzyme may increase achieved skeletal muscle uptake [63]. These findings led to the development of avalglucosidase alpha (neoGAA), a second-generation recombinant GAA replacement therapy with increased bis-M6P levels on the molecule.

The modification of neoGAA aims to increase receptor-mediated enzyme uptake and, hence, increase glycogen clearance and improve clinical efficacy [61]. In the recent Phase III COMET trial, neoGAA was demonstrated to achieve greater improvements in upright FVC and 6 min walk test in patients with LOPD as compared to alglucosidase alpha. Moreover, neoGAA exhibited a more favorable safety profile as well [64].

Besides pharmaceuticals, gene therapy is another area with huge potential for developing effective therapies for PD. Previous trials by Zhang et al. (2012) demonstrated partial biochemical correction of the skeletal muscles and diaphragm, leading to improved motor function in a mice PD model [65]. Furthermore, a Phase I/II trial by Smith et al. (2013) on AAV-mediated GAA gene therapy in children with PD showed that the patients’ unassisted tidal volume was significantly larger, and patient tolerance for the duration of unassisted breathing was increased as well. The results also suggested that the gene therapy applied was safe [66]. Nonetheless, further trials with more patients with larger diversity are needed to confirm the clinical benefits of gene therapy in managing the disease.

Early initiation of treatment is the key to optimizing the overall health outcomes in patients with PD, particularly IOPD. Nonetheless, the rarity of PD and the wide variety of clinical presentations hamper early diagnosis and subsequent treatment for the patients. Professional training on PD as well as other rare inherited diseases for frontline clinicians would enhance awareness of these diseases. Practically, the implementation of NBS would facilitate early identification of the PD patients, whereas ERT with alglucosidase alpha has been demonstrated to control PD effectively with a favorable safety profile. Certainly, further investigations on diagnostic tools and therapeutics for PD would provide insights on prompt diagnosis and proper treatments for the disease and, thus, are highly desirable.

## Figures and Tables

**Figure 1 children-09-01404-f001:**
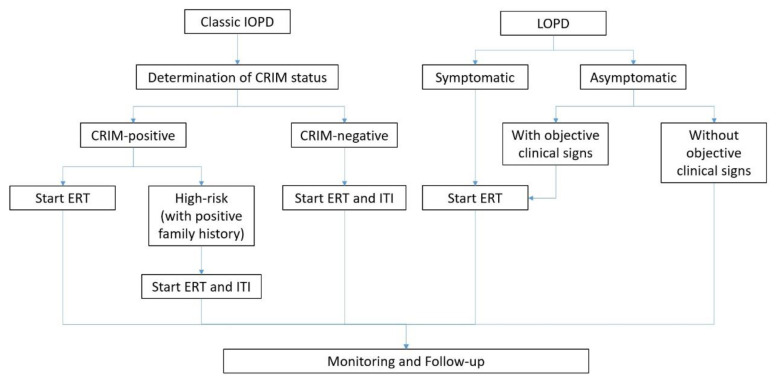
ERT treatment flowchart of PD.

**Table 1 children-09-01404-t001:** Common clinical presentations of PD.

IOPD	LOPD
Musculoskeletal Progressive muscle weakness Hypotonia Motor delay Macroglossia Reduced reflexes Heart Cardiomegaly Left ventricle hypertrophy Lungs Progressive respiratory symptoms Respiratory infections Others Difficulty in swallowing, eating and breastfeeding Psychomotor development delay Hepatomegaly	Musculoskeletal Progressive muscle weakness Unstable walking Tiptoe Low back pain Reduced reflex Difficulties in climbing stairs Scapula alata Gowers sign (dystrophy as result of proximal muscle weakness) Psychomotor development delay Lordosis/scoliosis Lungs Respiratory insufficiency Orthopnea Sleep apnea Effort dyspnea Exercise intolerance Respiratory infections Others Difficulty in swallowing and eating Tongue atrophy Hepatomegaly Morning headache Night somnolence

**Table 2 children-09-01404-t002:** Symptoms and observations suggestive of potential PD [3,11,13].

Infantile Form	Late-Onset Form
Potential patient group(s) Infants Symptoms and observations Poor feeding/failure to thrive Muscle weakness Respiratory difficulty Cardiac problems Shortened PR interval with a broad and wide QRS complex Cardiomegaly Left ventricular outflow obstruction Cardiomyopathy	Potential patient group(s) Infants, children, juveniles, and adults Symptoms and observations Proximal muscular weakness Respiratory insufficiency Typically rare cardiac involvement

**Table 3 children-09-01404-t003:** Assessment tests on PD disease progression [26].

Outcomes	Assessments
Skeletal muscle strength	Manual muscle testing (MMT) Assessed based on Medical Research Council (MRC) grading scale, ranging from 0 to 5 Handheld dynamometry (HHD) Measured in Newtons (N) and maximum isometric contraction values
Vital capacity	Recommended frequency: every 3–6 months [60] Performed in the upright and supine positions Formal pulmonary function testing (PFT) should be performed for patients with diaphragmatic weakness
Functional exercise capacity	6 min walk test Performed indoors along a 30 m long, straight and flat corridor. Count the laps to estimate the distance covered. Record patient’s dyspnea and fatigue
Quality of life	The Short Form 36 (SF36) Health Survey Assessed the changes in QoL in both physical health and mental health domains
Hearing	Audiology assessment Performed at baseline and then as clinically indicated
Laboratory tests	Performed every 3 months in the first year of diagnosis, followed by monitoring every 6 months in the following years if stable on ERT CK Reduction in CK level indicates response to treatment GAA antibody titers High and sustained antibody titers (HSAT) correlate with poor response to ERT in IOPD

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
