# Peer review of "The Clinical Management of Pompe Disease: A Pediatric Perspective"

_children, 2022, doi:10.3390/children9091404_

Round 1

Reviewer 1 Report

The manuscript provides a comprehensive review on pathogenesis, clinical presentation, diagnosis and treatment of Pompe disease.

The paper is quite well written, easy to follow, enriched with 3 tables and 1 figure, which seem to be very useful for clinicians.

However, it is still a review paper.

I propose to add the Author's own observations / experiences regarding patients with Pompe disease.

Author Response

An extra section titled “8. Case Sharing on ERT for PD” is added in the manuscript, which outlined 5 of my previous cases of managing Pompe Disease.

Reviewer 2 Report

This article is very clear and very good review about Pompe disease. Altough I have a question abouth cardiomyopathy: cardiomyopathy is rare in LOPD. Currently there are evidence about this concept because PD juvenile, and adult-onset forms cardimyophaty is presente and cause impact. It is possible to review about digestive features .

Author Response

Given LOPD is generally associated with a wider range of age of onset and clinical symptoms, the statement of "cardiomyopathy is rare in LOPD" is removed.

Reviewer 3 Report

This review on clinical treatment in Pompe disease gives a nice overview for the interested neuromuscular and neurological community as well as the interested pediatrician. The references are well prepared and cited and the overview has a clear focus and partition.

I have some remarks of minor misleadings, that should be corrected before publishing:

1. Title: since the manuscript deals mainly with patients treatment under the age of 18, the title seems misleading since the focus is clearly pediatrician.

2. Table 1: several grammatical mistakes: ...absence of reflexes...reduced reflexes...difficulty in swallowing and eating

Gowers sign (not: signs): this ist not a result of an "extreme" muscle weakness but of slowly proximal leg weakness

tongue atrophy in adult PD is not mentioned

3. Table 2: there seem to be missing some end of sentences in this table

e.g. respiratory infections/difficulty in ???

e.g. shortened PR interval wit a broad ???

4. line 102 Muscle biopsy is not the diagnostic tool of choice if you have less invasive (blood spot test) options

5. line 150...fibroblast culture is not the gold standard for making the diagnosis for a clinician as long as similar results can be achieved with the blood spot test...please specify

6 line 232: this is a false statement. neoGAA is availible in Germany since august 2022, the paragraphs on this new treatment should be adjusted properly

Author Response

1. Title: since the manuscript deals mainly with patients treatment under the age of 18, the title seems misleading since the focus is clearly paediatrician.

The title is changed by including the scope “In a Pediatric Perspective”.

2. Table 1: several grammatical mistakes: ...absence of reflexes...reduced reflexes...difficulty in swallowing and eating Gowers sign (not: signs): this ist not a result of an "extreme" muscle weakness but of slowly proximal leg weakness tongue atrophy in adult PD is not mentioned

The wordings are amended accordingly, with “Tongue atrophy” included in “others” of LOPD.

3. Table 2: there seem to be missing some end of sentences in this table e.g. respiratory infections/difficulty in ??? e.g. shortened PR interval with a broad ???

The typos have been amended accordingly.

4. line 102 Muscle biopsy is not the diagnostic tool of choice if you have less invasive (blood spot test) options

The term “commonly used” is revised. Also, the sentence “Moreover, in view of the traumatic nature of muscle biopsy, less invasive options, such as the blood spot test, are preferred” is added at the end of the paragraph to highlight this fact.

5. line 150...fibroblast culture is not the gold standard for making the diagnosis for a clinician as long as similar results can be achieved with the blood spot test...please specify

The statement is changed to “Analysis of GAA activity in fibroblast cultures obtained from skin biopsy can be applied in PD diagnosis”

6 line 232: this is a false statement. neoGAA is availible in Germany since august 2022, the paragraphs on this new treatment should be adjusted properly

The claim of “only treatment available” has been removed. The statement is amended accordingly. Also, the outdated content about FDA and EMA approval on neoGAA in paragraph 2 of section 9 has been removed.

Round 2

Reviewer 1 Report

This is my second review of the paper authored by Marques JS.

I recommend to publish the manuscript in its actual form.

Congratulations!